# Atomic structure of PI3-kinase SH3 amyloid fibrils by cryo-electron microscopy

Christine Röder [1,2], Nicola Vettore [2,6], Lena N. Mangels[2,6], Lothar Gremer [1,2], Raimond B.G. Ravelli [3], Dieter Willbold [1,2], Wolfgang Hoyer [1,2], Alexander K. Buell [2,4] & Gunnar F. Schröder [1,5]

High resolution structural information on amyloid fibrils is crucial for the understanding of their formation mechanisms and for the rational design of amyloid inhibitors in the context of protein misfolding diseases. The Src-homology 3 domain of phosphatidyl-inositol-3-kinase (PI3K-SH3) is a model amyloid system that plays a pivotal role in our basic understanding of protein misfolding and aggregation. Here, we present the atomic model of the PI3K-SH3 amyloid fibril with a resolution determined to 3.4 Å by cryo-electron microscopy (cryo-EM). The fibril is composed of two intertwined protofilaments that create an interface spanning 13 residues from each monomer. The model comprises residues 1–77 out of 86 amino acids in total, with the missing residues located in the highly flexible C-terminus. The fibril structure allows us to rationalise the effects of chemically conservative point mutations as well as of the previously reported sequence perturbations on PI3K-SH3 fibril formation and growth.

[1] Institute of Complex Systems, Structural Biochemistry (ICS-6) and JuStruct, Jülich Center for Structural Biology, Forschungszentrum Jülich, 52425 Jülich, Germany. [2] Institut für Physikalische Biologie, Heinrich-Heine-Universität Düsseldorf, 40225 Düsseldorf, Germany. [3] The Maastricht Multimodal Molecular Imaging Institute, Maastricht University, Universiteitssingel 50, 6229 ER Maastricht, The Netherlands. [4] Department of Biotechnology and Biomedicine, Technical University of Denmark, Søltofts Plads, 2800 Kgs Lyngby, Denmark. [5] Physics Department, Heinrich-Heine-Universität Düsseldorf, 40225 Düsseldorf, Germany. [6] These authors contributed equally: Nicola Vettore, Lena N. Mangels. Correspondence and requests for materials should be addressed to A.K.B. (email: alebu@dtu.dk) or to G.F.S. (email: gu.schroeder@fz-juelich.de)

Alzheimer's and Parkinson's Disease as well as spongiform encephalopathies are prominent examples of protein misfolding diseases[1]. These disorders are characterised by the presence of amyloid fibrils[2]. Amyloid fibrils are straight and unbranched thread-like homopolymeric protein assemblies, which are stabilised by backbone hydrogen bonding between individual peptide molecules. These interactions lead to a highly ordered, repetitive cross-β architecture, in which the β-strands run perpendicularly to the fibril axis.

It has been shown that in the case of neurodegenerative protein misfolding diseases, the final amyloid fibrils are often not the most cytotoxic species, but that small, oligomeric precursors are more hydrophobic and more mobile and hence more prone to deleterious interactions with cellular components[3]. However, recent progress in the mechanistic understanding of amyloid fibril formation shows that the mature fibrils can be the main source of toxic oligomers, due to their role as catalytic sites in secondary nucleation processes[4]. Furthermore, in the case of systemic amyloidosis diseases, where amyloid fibrils form in organs other than the brain, the amyloid fibrils themselves are the deleterious species, as their presence in large quantities can disrupt organ functions[5].

Until recently, structural information on amyloid fibrils could only be obtained from relatively low-resolution methods, such as X-ray fibre diffraction[6], limited proteolysis[7] and H/D exchange[8]. High-resolution structural information on amyloid fibrils has only become available in recent years through progress in solid state NMR spectroscopy (ssNMR)[9–11] and cryo-electron microscopy (cryo-EM). In particular cryo-EM enables atomic resolution structures of amyloid fibrils to be determined, and this has indeed been achieved in a few cases so far[12–23]. Such detailed structural information is crucial for the understanding of amyloid formation mechanisms, as well as for the rational design of inhibitors of the individual mechanistic steps, such as fibril nucleation and growth[24].

Here we present the high-resolution cryo-EM structure of amyloid fibrils of the Src-homology 3 domain of phosphatidyl-inositol-3-kinase (PI3K-SH3). SH3 domains are kinase sub-domains of usually <100 amino acids length and have been found to be part of more than 350 proteins, ranging from kinases and GTPases to adaptor and structural proteins, within various organisms[25]. SH3 domains are known to play a significant role in several signalling pathways where they mediate protein-protein interactions by recognising PxxP sequence motifs[26,27]. The structure of natively folded PI3K-SH3, a domain consisting of 86 amino acids from bovine PI3K, has been well-characterised by X-ray crystallography and NMR spectroscopy[26,28].

Used initially as a model system for protein folding studies[29], PI3K-SH3 was among the first proteins discovered to form amyloid fibrils in the test tube, while not being associated to any known human disease[30]. Fibril formation was observed at acidic pH, where in contrast to the native fold at neutral pH[27,28], monomeric PI3K-SH3 lacks a well-defined secondary structure[30–32]. Since this discovery, PI3K-SH3 has played a pivotal role in advancing our fundamental understanding of the relationships between protein folding, misfolding and aggregation. Indeed, the hypothesis of the amyloid fibril as the most generic 'fold'[1] that a polypeptide can adopt was significantly shaped by the finding that PI3K-SH3 forms amyloid fibrils. Many pioneering studies on the basic biochemical, structural and mechanistic features of amyloid fibrils have been performed with PI3K-SH3. Early cryo-EM measurements highlighted the need for conformational rearrangement of the sequence within the fibril[33]. The important role of the destabilisation of native secondary structure elements and the need for non-native contacts and extensive structural rearrangements during the formation of

fibrillar aggregates was also observed for a related SH3 domain[34]. Despite not being related to any human disease, PI3K-SH3 aggregates were shown to be cytotoxic, suggesting sequence-independent toxic properties of amyloid fibrils and their precursors[35]. PI3K-SH3 also provided insight into the kinetics of molecular recycling of the monomeric building blocks of the fibril[36], as well as into the dynamics of the formation of oligomeric precursors of amyloid fibrils[37].

The structure of PI3K-SH3 fibrils we present here is in agreement with previous ssNMR data[38], but we find that its interface is orthogonal to that suggested previously based on a low resolution reconstruction[33]. Indeed, the inter-filament interface in PI3K-SH3 fibrils is large compared to that of other amyloid fibrils determined to-date, and is formed from residues distant in primary sequence. With the atomic model we can rationalise the effect of newly designed, as well as previously reported sequence variants of PI3K-SH3 on the kinetics of fibril growth. Our study therefore not only adds important insight into the structural variety of amyloid fibrils, but also demonstrates how such structures can be used to rationalise the dynamics of protein assembly processes.

## Results

**Structure determination by cryo-EM.** Fibril formation by the full-length PI3K-SH3 domain under acidic solution conditions[30] led to long, straight fibrils of which the main population could be structurally determined by cryo-EM (Fig. 1). High overall homogeneity of the preparation has been shown by atomic force microscopy (AFM) and negative staining EM imaging (Fig. 2, Supplementary Figs. 1 and 2). Nevertheless, two different morphologies could be distinguished in both AFM and EM images (Fig. 2, Supplementary Fig. 1). The predominant, thick morphology has a diameter of 7–8 nm while the other, thinner morphology exhibits about half the diameter of the thick fibril (Supplementary Fig. 2).

For cryo-EM imaging, samples of PI3K-SH3 amyloid fibrils were flash-frozen on Quantifoil cryo-EM grids and imaged with a Tecnai Arctica microscope (200 kV) equipped with a Falcon 3 direct electron detector (Supplementary Fig. 3). Image processing and helical reconstruction were performed with RELION-2[39–41]. A three-dimensional density map for the thick PI3K-SH3 fibril could be reconstructed to an overall resolution of 3.4 Å. The clear density of the fibril allowed us to build an atomic model for residues 1–77 out of a total of 86 amino acids (Fig. 1). The missing nine residues are located at the C-terminus, which shows blurred density likely due to substantial flexibility. Previously reported low-resolution cryo-EM[33] data are in good agreement with our structure (Supplementary Fig. 4).

**Architecture of the PI3K-SH3 amyloid fibril.** The thick PI3K-SH3 fibril is a left-handed helical structure consisting of two intertwined protofilaments, and is thus called DF (double filament) fibril. The handedness of the density reconstruction was determined by comparison with AFM images (see Methods and Supplementary Fig. 5). From an analysis of the fibril height profiles in AFM images, we determined the helical pitch to be $170 \pm 10$ nm in reasonable agreement with a pitch of 140 nm obtained from the cryo-EM structure.

Protofilament subunits (PI3K-SH3 monomers) are stacked in a parallel, in-register cross-β structure. The spacing between the layers of the cross-β structure is around 4.7 Å and well visible in the density (Supplementary Fig. 6). The subunits in the two opposing protofilaments are not on the same z-position along the fibril axis but are arranged in a staggered fashion (Fig. 1c). The helical symmetry is therefore described by a twist of 179.4 ° and

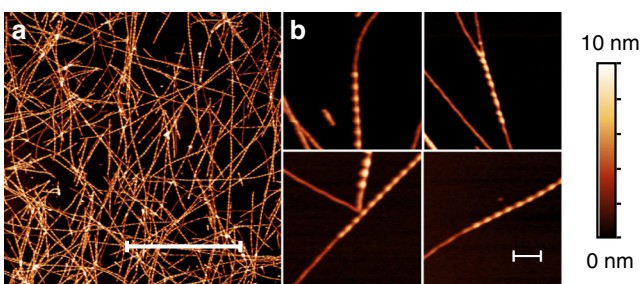

**Fig. 1** Double Filament SH3 fibril structure. **a** Cross-section of the double filament (DF) PI3K-SH3 fibril (two monomers). The different protofilaments are colored blue and orange. Density maps are shown at a contour level of 1.4 σ. The density map was sufficiently clear to model residues 1–77. **b** Side view of filaments twisting around each other displaying a total of 125 layers. **c** Relative arrangement of two adjacent PI3K-SH3 monomers within the fibril, showing a 2.35 Å shift between the protofilaments

**Fig. 2** AFM images of PI3K-SH3 amyloid fibrils. **a** AFM image of PI3K-SH3 fibrils (scale bar, 3 μm). **b** Four different close-up views of fibrils showing both the thick, double filament (DF) fibril and thin, single filament (SF) fibril morphology (scale bar, 300 nm)

rise of 2.35 Å, which corresponds to an approximate $2_1$-screw symmetry. The same staggering arrangement has been observed previously for other fibril structures, such as amyloid-β(1–42)[13] and paired-helical filaments of tau[23].

Each protofilament subunit, or PI3K-SH3 monomer, consists of seven parallel in-register β-strands (β1, aa1–5; β2, aa7–19; β3, aa22–26; β4, aa28–34; β5, aa46–56; β6, aa59–68; β7, aa72–77) that

are interrupted by either sharp kinks, glycine or proline residues —or a combination of those (Figs. 1a and 3, Supplementary Fig. 6). In particular glycine residues at kink positions have also been observed in other amyloid structures[17]. The total of seven kinks and turns (Fig. 3a) results in an amyloid key topology[20], which includes a structural motif similar to the bent β-arch described by Li et al.[17] for the α-synuclein fibril.

By encompassing 13 residues of each monomer, 26 in total (Supplementary Fig. 7), the inter-filament interface of the PI3K-SH3 DF fibril is very large. The protofilaments mainly interact through two identical symmetry-related hydrophobic patches (Fig. 4) at the periphery of their interface, between the bottom part of the bent β-arch motif and the respective C-terminal part of the opposing protofilament. The stability is provided by the hydrophobic clusters of $Val^{38}$, $Leu^{40}$, $Gly^{41}$, $Phe^{42}$, $Val^{74'}$, $Tyr^{67'}$ and $Val^{38'}$, $Leu^{40'}$, $Gly^{41'}$, $Phe^{42'}$, $Val^{74}$, $Tyr^{67}$ (Figs. 1a and 4).

The amyloid-characteristic cross-β motif composed of parallel in-register β-sheets connects the different DF fibril layers and therefore contributes the largest share of intermolecular contacts (Fig. 5, Supplementary Fig. 6). This cross-β stacking is complemented by multiple inter- and intramolecular contacts including side chain interactions in homosteric and heterosteric zippers[42]. A further noteworthy feature of the structure is the fact that the PI3K-SH3 subunits are not planar but extent along

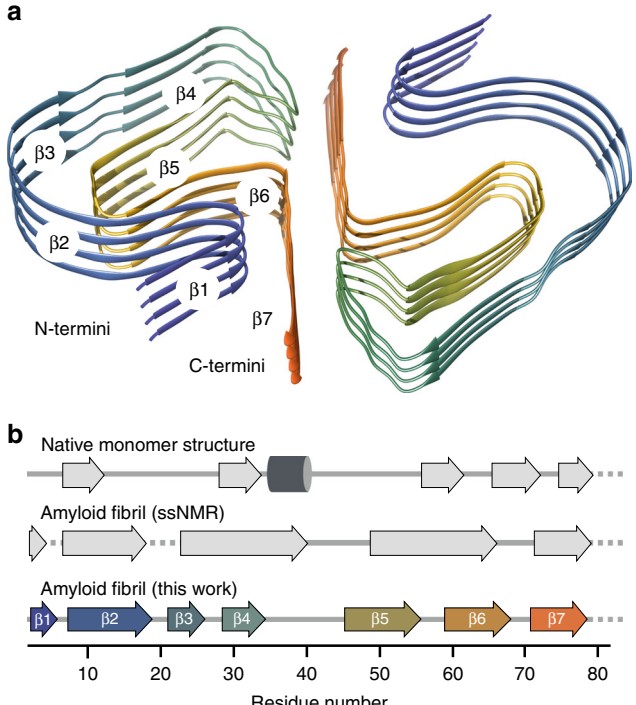

**a**

β4
β3
β5
β6
β2
β1
β7
N-termini
C-termini

**b**

Native monomer structure

Amyloid fibril (ssNMR)

Amyloid fibril (this work)

β1 β2 β3 β4 β5 β6 β7

10  20  30  40  50  60  70  80

Residue number

**Fig. 3** Secondary structure comparison of the PI3K-SH3 DF fibril. Secondary structure of the presented cryo-EM structure compared to data obtained previously by solid-state NMR (fibrils) or liquid-state NMR (monomeric, native state). **a** Tilted cross-section of four SH3 DF fibril layers. Secondary structure is predominantly formed by seven cross-β sheets. **b** Secondary structure comparison to the native solution structure (PDB: 1PKS[28]) and ssNMR results (fibrils) modified according to Bayro et al.[38]. Flexible regions are shown as dashed lines, β-sheets as arrows, and the helix as a cylinder

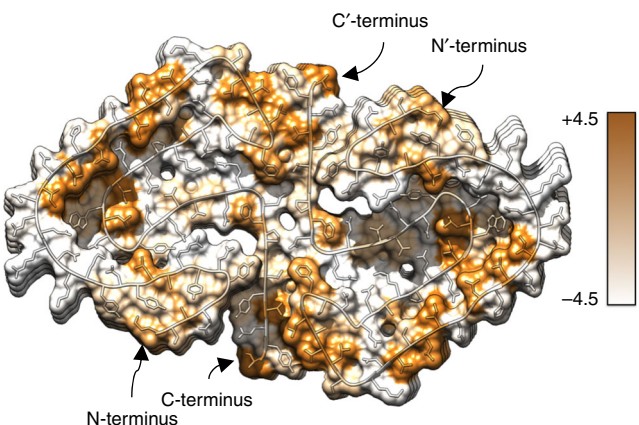

C′-terminus
N′-terminus
+4.5
−4.5
C-terminus
N-terminus

**Fig. 4** Hydrophobicity of the fibril cross-section. Hydrophobicity levels of the SH3 DF fibril cross-section are coloured according to Kyte-Doolittle[61]. Hydrophobic residues are mainly packed within the fibril core, while hydrophilic residues point towards the solvent. Hydrophobic patches in both monomers are clearly visible and are spanning several β-sheets

the fibril axis (Fig. 5). The subunits within a protofilament therefore interact not only with the layer directly above $(i + 2)$ and below $(i − 2)$, but also with layers up to $(i + 6)$ and $(i − 6)$; the subunits are interlocked.

A single PI3K-SH3 monomer in the DF fibril exhibits an amyloid key topology (Fig. 1a), which is stabilised mainly by hydrophobic patches stretching between the strands β3–β5 (Ile$^{22}$ $(i)$, Trp$^{55}$ $(i − 4)$, Leu$^{24}(i)$, Leu$^{26}(i)$) (Fig. 4), and hydrogen bonds

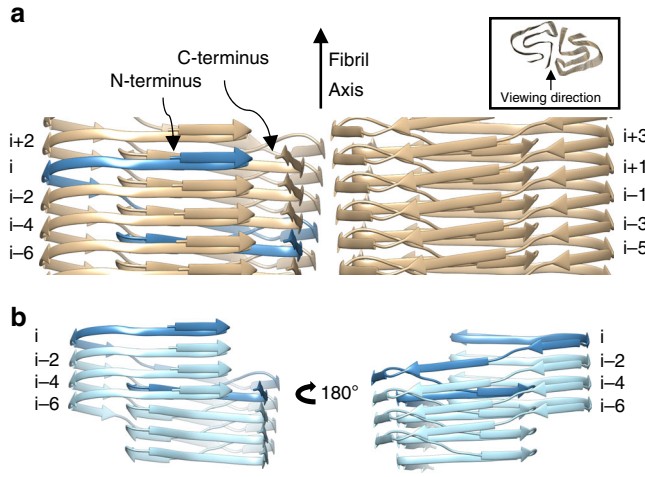

**a**

C-terminus
N-terminus
Fibril Axis
Viewing direction

i+2
i
i−2
i−4
i−6

i+3
i+1
i−1
i−3
i−5

**b**

i
i−2
i−4
i−6

180°

i
i−2
i−4
i−6

**Fig. 5** Side view of the secondary structure of the atomic model. **a** Single subunit $i$ highlighted in blue with adjacent subunits in beige, described by even numbers, while the subunits of the opposite protofilament are described by odd numbers. **b** View of the minimal fibril unit $(i)$ to $(i$-6$)$ in one filament. The minimal fibril unit is displayed from two views by a turn of 180° highlighting the course of one monomer spanning several fibril layers

or salt bridges between strands β5 and β6 (Asp$^{44}(i)$, Arg$^{66}(i − 2)$, Glu$^{47}(i)$) and β2–β6 (Arg$^{9}(i)$, Glu$^{61}(i − 6)$) (Figs. 1a and 3a). In the turn between strands β5 and β6, Asp$^{13}(i + 4)$ might bind to Lys$^{15}(i + 4)$ while Glu$^{17}(i + 4)$ might interact with Asn$^{57}(i)$ (Figs. 1a and 3a). The bent β-arch motif between strands β4 and β5 is potentially strengthened by a contact between Asn$^{33}(i)$ and Gln$^{46}(i)$ that would tie the motif together (Figs. 1a and 3a). Further possible electrostatic interactions can be observed between strands β2 and β6 where the amino-group of Arg$^{9}$ $(i+4)$ might exhibit a salt bridge to Glu$^{61}(i$-2$)$ (Figs. 1a and 3a). In addition, aromatic side chains are located in close proximity to glycine residues Tyr$^{6}(i)$-Gly$^{71}(i + 6)$, Phe$^{42}(i)$-Gly$^{67}(i + 2)$, Tyr$^{73}$ $(i)$-Gly$^{5}(i − 6)$, indicating a potential involvement of glycine-aromatic Cα-H⋯π-interactions[43,44]. Further aliphatic-aliphatic and aromatic-aliphatic interactions comprise Ile$^{29}(i)$-Pro$^{50}(i − 2, i − 4)$-Ala$^{48}(i − 2, i − 4)$, Met$^{1}(i)$-Tyr$^{12}(i)$, and Leu$^{11}(i)$-Tyr$^{59}$ $(i$-4$)$. We could also observe possible interactions in-between monomer layers, so-called hydrogen bond ladders, e.g. with Gln$^{7}$, Gln$^{46}$, Asn$^{57}$ and Asn$^{60}$. Additionally, these ladders could potentially be formed as well by side chains pointing outside of the fibril such as Glu$^{4}$, Glu$^{20}$, Asp$^{21}$, Asp$^{23}$ and Asp$^{28}$. Remarkably, nearly all side chains in the outer Cα-chain that are oriented towards the solvent are non-hydrophobic (Fig. 1a) (Ser$^{0}$-Glu$^{4}$, Tyr$^{14}$-Ser$^{36}$). Hence, the polar outside of the fibril shields the hydrophobic interface (Fig. 4). Only the hydrophobic residues Leu$^{30}$ and Val$^{32}$, which are located on the fibril surface next to the bent β-arch motif, disrupt this pattern (Figs. 1a and 4).

Further hints towards the role of electrostatic interactions in maintaining the structure, comes from a pH-shift experiment. By changing the pH from 2 to 7.4, we observed that fibrils depolymerise almost completely after 1 h (Supplementary Fig. 8), in agreement with the highly dynamic nature of PI3K-SH3 fibrils reported previously[36]. We propose three clusters that may be highly influenced by this pH shift: (a) the space between strands β5 and β6 (Asp$^{44}(i)$, Arg$^{66}(i − 2)$, Glu$^{47}(i)$); (b) the interactions between Asp$^{13}(i)$ and Lys$^{15}(i)$, and between Glu$^{17}(i + 4)$ and Asn$^{57}(i)$; (c) the solvent-exposed and protonatable patch involving Asp$^{21}(i)$, Asp$^{23}(i)$ and His$^{25}(i)$ (also highlighted as having a fundamental role in amyloid formation[45]).

The secondary structure of PI3K-SH3 fibrils and monomers has been analysed by comparing available solution NMR

(monomeric native fold, PDB: 1PKS)[28] and ssNMR data (amyloid fold)[38] with our model (Fig. 3b). The only feature that is shared by all models is the flexible C-terminus starting around residue 80. Analysis of protein contact maps via the Contact Map WebViewer[46] of the DF fibril compared to the monomeric native structure[28] showed no consistent residue contacts in both structures, illustrating the substantially different conformation that the monomer unit has to adopt in order to incorporate into the fibril. The secondary structure of the native and amyloid fold differs substantially apart from a β-sheet between residues 70–80. While the monomeric native structure is characterised by one helix between residues 34 and 39, β-sheets and multiple flexible loop-regions, the DF fibril consists of seven β-sheets exclusively, that are almost uninterrupted (Fig. 3b). The longest break in the β-sheet pattern of the fibril is the bent β-arch motif leading to a rigid loop between Gly[35] and Gly[45] (Figs. 1a and 3a). Our findings are consistent with former results by Bayro et al.[38] who proposed a PI3K-SH3 amyloid model based on solid-state NMR data. Both structures show β-sheets as the only secondary structure motif with most of the β-sheet regions corresponding (Fig. 3b). The main differences between the ssNMR and cryo-EM structures are found in the region between residues 25 and 60. Here, ssNMR data suggest two wide-spanning β-sheets while according to cryo-EM data this region consists of not two but four β-sheets, that are disrupted by a glycine residue, Gly[27] (β3–β4) (Supplementary Fig. 6) and a sharp kink, Gly[54]-Tyr[59] (β5–β6) (Fig. 3a).

**Impact of mutations on the SH3 aggregation**. We probed the sensitivity of the fibril growth kinetics towards chemically conservative single point mutations by substituting isoleucine residues for alanine across the protein sequence. We expressed and purified the sequence variants and experimentally quantified the rates at which five different variants (I22A, I29A, I53A, I77A, I82A) elongated wildtype (WT) fibrils by quartz crystal microbalance (QCM) measurements (Supplementary Fig. 9)[47,48]. This technology is ideally suited for such cross-seeding experiments. The fibril growth rates of WT and variant proteins can be directly compared, given that the same, constant ensemble of fibrils is monitored. An additional big advantage is that the use of WT seeds as templates ensures that the sequence variants adopt the same fibril structure as the WT, and therefore the change in fibril growth kinetics with respect to the wild type can be interpreted in terms of the perturbation induced by the sequence modification.

We measured the rates of WT fibril elongation by the different variants and expressed the rates relative to that of the elongation by WT monomer (Fig. 6, Supplementary Fig. 9). We found that the relative elongation rates differ by more than two orders of magnitude, with the mutations in the first third of the sequence, as well as around the middle of the sequence, displaying slow elongation rates (I22A = 0.02 ± 10.9%, I29A = 0.02 ± 7.5%, I53A = 0.003 ± 72.7%), whereas the mutations close to or within the disordered C-terminus display approximately the same rates as the wild type (I77A = 1.13 ± 9.3%, I82A = 1.59 ± 10.3%). We also took AFM images of the different single point mutants at a concentration of 100 μM that were all seeded with WT-derived fibrils and incubated for 2 days at room temperature under quiescent conditions (Supplementary Fig. 10). From these images, it can be seen that all fibrils at the end of the experiment have a very similar morphology and length, except for those in the sample with I53A, where shorter fibrils are observed. The fibril growth of all the mutants seems to have come to completion within this time scale leading to very similar fibril lengths. Only in the case of I53A, the fibril growth rate is so slow that the available monomer was only partly incorporated into the seed fibrils

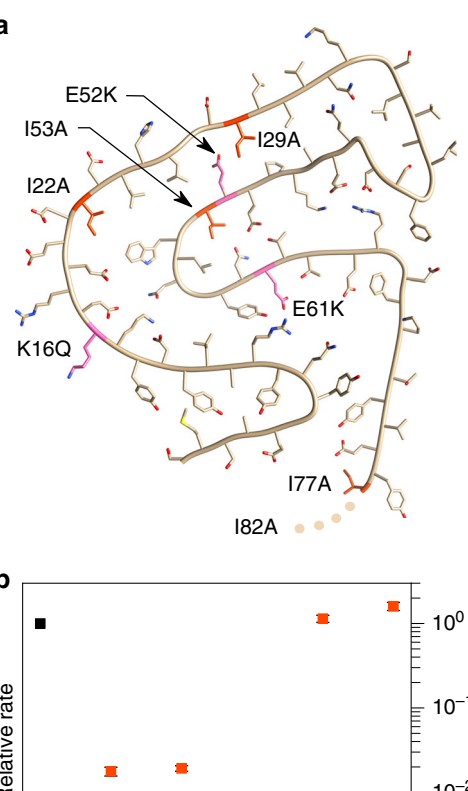

**Fig. 6** Point mutations and their effect on fibril elongation rates. **a** Top view of a PI3K-SH3 monomer derived from DF fibrils. Highlighted residues: Isoleucine residues mutated to alanine (I22A, I29A, I53A, I77A, I82A), from this work (red) and charged residues mutated by Buell et al.[47] (K16Q, E52K, E61K, pink). Mutation I82A is missing in the model due to the flexible C-terminus. **b** Elongation rate of Ile-to-Ala mutants. The elongation rates are normalised to a WT rate of one. Source data are provided as a Source Data file

during the course of the experiment. However, the very high mass sensitivity of the QCM permits to resolve even the growth rate of this slow growing mutant, and hence we base our analysis of the relative growth rates exclusively on the QCM experiments (Supplementary Fig. 9).

## Discussion

Concerning fibril architecture and polymorphism, we observed that, in addition to the thick 7–8 nm SH3 DF fibril consisting of two filaments, also thin fibrils of about half the diameter are present in both AFM (Fig. 2, Supplementary Fig. 2) and negative stain EM images (Supplementary Fig. 1). We therefore hypothesise that the thin fibrils correspond to single filament fibrils (SF fibrils) of the same type that make up the DF fibrils. Our hypothesis is further supported by AFM images that suggest that long SF fibrils can contain stretches that appear to be identical to DF fibrils (Supplementary Fig. 2). It therefore seems likely that fragments of an additional protofilament can attach to or grow on a given SF fibril, to form DF fibrils. The incomplete cooperativity between the elongation of the individual filaments in a DF fibril suggests relatively weak interactions between the filaments. This, in turn, implies that one filament could well be stable without

contact to another filament. We conclude that, at least in the case of PI3K-SH3, inter-filament interface contacts are not necessary for fibril formation.

A similar observation has been made for β2-microglobulin fibrils[15], which can comprise a single protofilament as well as two or more filaments resulting in at least six different polymorphs without major differences in the filament structure. Accordingly, a mixture of all polymorphs yielded only a single set of NMR resonances[15]. The stability of individual PI3K-SH3 filaments is particularly noteworthy, inasmuch as the inter-filament interface in the DF fibrils is comparatively large (Supplementary Fig. 7). A large inter-filament interface does therefore not indicate stable inter-filament interactions. We note here also that the identical conformation of every subunit along the fibril axis is in line with fibril growth by PI3K-SH3 monomer addition[49,50].

A single subunit in the PI3K-SH3 DF fibril is not planar but winds itself along the protofilament axis (Fig. 5), which leads to interlocking within a protofilament. In contrast, in a planar subunit, the inter-subunit interactions within one protofilament would consist exclusively of the cross-β pattern, while all transverse interactions would be intramolecular. Here, however, one subunit within one protofilament is not only in contact with its direct neighbours (above and below) but with four other monomers in total, i.e. in addition to the longitudinal hydrogen bonds in the cross-β structure, there are other transverse interactions between the subunits. This staggered arrangement leads to an interlocking mechanism connecting several monomers within a protofilament, which very likely further stabilises the structure (Fig. 5). Interestingly, this interlocking mechanism is commonly seen in other amyloid fibril structures determined by cryo-EM[13,17,18,20,21]. It should be noted that NMR data can only distinguish between intra- and intermolecular contacts but cannot directly reveal a potential staggering of subunits along the fibril axis.

Given that the interlocking is observed in other fibril structures as well, it might in general contribute to the formation of stable fibrils by optimisation of the side chain packing[10]. In addition, the staggered architecture might in part be responsible for the templating effect during fibril elongation, as it establishes a rugged binding interface that may guide the incoming monomer into the fibril conformation, engaging it in more intermolecular contacts than a flat interface could.

As a measure for the interlocking of subunits we have previously defined the concept of a minimal fibril unit[13], which is the smallest fibril structure fragment in which the capping subunits at both ends would have established the same full contact interface with other constituting monomers as the capping subunits of an extended fibril. Since we hypothesise that one protofilament of the DF fibril could exist on its own and is identical to the SF fibril, here we describe the minimal fibril unit also for a single protofilament. For the PI3K-SH3 fibril, the minimal fibril unit has a size of four subunits when considering an individual protofilament, and a size of eight subunits in the case of the DF fibril (Fig. 5b).

In order to rationalise kinetic data of fibril formation and growth, we substituted five different isoleucine residues with alanines, probing how these chemically conservative single point mutations at different positions affect the addition of new monomers to the fibrillar structure formed by the wild type sequence. Three mutations (I22A, I29A, I53A) showed a strong decrease of the elongation rate of two to three orders of magnitude compared to the wild type sequence (Fig. 6). In these positions, the side chains of the three isoleucines are all pointing towards the fibrillar core (Fig. 6). While the chemical nature of the amino acid substitution we chose is conservative (aliphatic to aliphatic), the bulkiness of the side chain decreases. The ability of

these variants to elongate the WT structure, albeit significantly slower than the WT sequence itself, suggests that the formation of a cavity due to the reduced bulkiness is energetically tolerated. However, the free energy difference between the monomeric state and the transition state (structural ensemble) of the elongation reaction appears to be increased. Such an increase in energy difference could come either from a stabilisation of the monomeric state, or from a destabilisation of the transition state. The former seems less likely, due to the mostly disordered nature of PI3K-SH3 at pH 2, while the latter possibility could be caused by the reduction in hydrophobic contacts between the monomer and the fibril end. This conclusion, which assumes some degree of contact between the monomer and the fibril end in the transition state for fibril growth, is in excellent agreement with previous results that underline the importance of the sequence hydrophobicity for the magnitude of the elongation free energy barrier[51]. The remaining two mutations (I77A, I82A) (Fig. 6) are instead located close to or within the flexible C-terminus and show indeed a much weaker or no effect on the elongation rate. In both cases a chemical modification to alanine does not perturb any interactions crucial for the energetics of the transition state.

The availability of the high-resolution structure also allows us to rationalise the influence of previously reported single point mutations of PI3K-SH3 on the kinetics of fibril elongation. In a previous study[47], the effect of changes in charge at three different positions (Fig. 6; K16Q, E52K, E61K; residues depicted in pink) led to very different effects on the elongation rate. While these mutations, similar to the ones we have designed and studied in the present work, could lead to different fibrillar structures if induced to form fibrils de novo, the use of WT seed fibrils in both studies allows us to discuss here the effect of these mutations in the light of the present fibril structure. This is because the well-known templating effect in amyloid fibril growth imprints the structure of the fibril template onto the monomeric protein that adds onto the fibril end. The mutation E52K probably leads to the creation of a positive charge inside the hydrophobic fibril core, a highly unstable arrangement. In the WT fibril, Glu[61] can form a salt bridge with Arg[9], which leads to an additional driving force for the deprotonation of Glu[61]. If Glu[61] is substituted by a lysine (E61K), two positive charges come into close proximity, again leading to a highly unstable situation. On the other hand, the mutation K16Q does not lead to any major change in the kinetics of fibril elongation, which is most likely due to the side chain pointing towards the outside of the fibril[52].

Our structural model can also help to understand the effects of further sets of previously investigated sequence changes[45]. With the aim of understanding which part of the sequence plays a major role in the amyloidogenicity of the protein, different portions of the sequence were replaced or mutated. In the vast majority of cases, the amyloidogenic behaviour was completely abolished, which we can now explain with our structural model: the introduction of bulky or charged residues facing the inner core of the structure destabilises the present amyloid fold, as evidenced by the mutants E17R/D23R and Q7E/R9K/E17R/D23R[45]. The only mutation that does not show a significant decrease in amyloidogenicity does not involve changing the charge of a buried residue: the mutant referred to as PI3-QMR (E17Q/D23M/H25R) modifies a single charge of outward pointing residue 25 (assuming Glu[17] and Asp[23] to be protonated at pH 2[45]).

It should also be noted that the circularisation of the PI3K-SH3 sequence through the use of disulphide bridges causes a decrease of the elongation rate but does not prevent the circularised mutant to acquire the amyloid conformation[53]. The close proximity of the N- and C-termini (Fig. 1a), in combination with the flexibility of the C-terminal nine residues, likely allows the

cyclised (disulfide bridge between Cys[3] and Cys[82]) sequence to form an amyloid structure very similar, if not identical to the one presented here.

A noteworthy example of substantial sequence modification is the grafting of the N-Src loop of SPC-SH3 onto PI3K-SH3, while simultaneously removing the stretch from residue 31 to 53 from PI3K-SH3[54]. This operation did not remove the amyloidogenic properties from the modified PI3K-SH3 domain. The removed sequence stretch is part of the bent β-arch motif in the amyloid conformation. The removal of this prominent motif could have been expected to lead to a more significant impairment of amyloid fibril formation. However, the altered sequence is likely to be able to respond to this strong perturbation by forming an alternative structure.

In summary, we have determined the structure of a PI3K-SH3 amyloid fibril. The PI3K-SH3 fibril has been extensively studied in the past and the effect of many mutations on the kinetics of amyloid formation has been described. The atomic structure of the PI3K-SH3 fibril enables us now to rationalise the effect of these mutations, which is the basis for understanding the sequence-dependence of amyloidogenicity and ultimately the determinants of amyloid formation in general.

## Methods

**Protein production**. WT and mutants of the bovine PI3K-SH3 domain were purified according to the protocol of Zurdo et al.[31]. All constructs contain a 6xHis-tag linked to the protein by a thrombin cleavage site. The sequence of the WT protein after cleavage is the following, with the peptide Gly-Ser remaining as overhang from the cleavage:

```
GSMSAEGYQYRALYDYKKEREEDIDLHLGDILTVNKGSL
VALGFSDGQEAKPEEIGWLNGYNETTGERGDFPGTYVEYIGRK
KISP
```

The protein was expressed in a BL21 E. coli strain with TB medium for auto-induction containing 0.012 % glucose and 0.048 % lactose. The cells were grown for over 24 h and then harvested by centrifugation. After resuspension in sodium phosphate buffer (50 mM sodium phosphate pH 8, 5 mM imidazole and 100 mM NaCl), the cells were disrupted by sonication, in presence of protease inhibitors and DNAse. The lysate was centrifuged, and the supernatant loaded on a Ni-NTA Superflow Cartridge (Qiagen, Venlo, Netherlands) equilibrated in 50 mM sodium phosphate pH 8, 5 mM imidazole and 100 mM NaCl. The protein was eluted with a linear gradient from 5 to 300 mM imidazole in 50 mM sodium phosphate pH 8, 100 mM NaCl in 25 ml elution volume. Fractions containing the protein were collected and cleaved overnight at 7 °C with 1 unit of thrombin (from bovine plasma, Sigma-Aldrich Saint Louis, Missouri, USA) per 1 mg of protein. The cleaved solution was then concentrated and loaded on a SEC HiLoad 26/60 Superdex 75 column (GE Healthcare, Chicago, Illinois, USA) equilibrated with 5 mM ammonium acetate pH 7. Fractions containing the PI3K-SH3 domain were collected and lyophilised for further use.

**Fibril formation**. The lyophilised protein was resuspended in 10 mM glycine-hydrochloride pH 2.5 buffer at a final concentration of ca. 200 μM. The solution was shaken in an Eppendorf tube at 1400 rpm at 42 °C for 24 h to form seeds. These seeds were then sonicated in an Eppendorf tube in a volume of ca. 500 μl for 15 s (1 s 'on', 2 s 'off', 10 % amplitude) with a Bandelin Sonopuls using a M72 probe. To prepare the twisted fibrils, a new solution with ca. 100 μM monomer in 10 mM glycine-hydrochloride pH 2.5 was then mixed with 5 μM of equivalent seeds mass and incubated without stirring at 50 °C overnight.

**AFM imaging**. The fibril samples were diluted in 10 mM glycine-hydrochloride, pH 2.5 to a concentration of 5 μM and 10 μl were pipetted on a mica substrate. After 10 min of incubation, the mica was washed extensively with milliQ water and dried under a nitrogen gas flush. The pictures were taken in tapping mode on a Bruker Multimode 8 (Billerica, Massachusetts, USA) using OMCL-AC160TS cantilevers (Shinjuku, Tokyo, Japan).

**Fibril elongation measurements with QCM**. The elongation rate of PI3K-SH3 fibrils was measured through immobilisation of fibrils on a QCM sensor and subsequent incubation with monomer solution[50]. To immobilise the fibrils on the sensor, chemical modification is necessary. To achieve that, the fibrils were mixed at a final concentration of 50 μM in buffer (10 mM glycine-hydrochloride, pH 2) with EDC (1-ethyl-3-(3-dimethylaminopropyl)carbodiimide hydrochloride) (1 M) and cystamine hydrochloride (0.5 mg ml⁻¹). After pelleting and washing the chemically modified fibrils, they were sonicated in an Eppendorf tube in a volume of ca. 500 microlitre with a Bandelin Sonopuls using a MS72 probe (10%

amplitude, 15 s, 1 s 'on', 2 s 'off'). The gold sensors (Biolin Scientific, Gothenburg, Sweden) were then incubated with the above-mentioned solution overnight in a 100% humidity environment. The measurements were performed with a QSense Pro (Biolin Scientific, Gothenburg, Sweden) by measuring the elongation rate as change in resonant frequency over time. With the temperature set at 25 °C, the monomer solutions were injected for 30 s at a flow rate of 100 μl per second and the measurement lasted until a stable slope was reached. To obtain the relative rates, the protein solutions were injected in different sensor chambers after a WT injection, the latter being used as normalization reference. Two different triplicate measurements were performed for I53A. Two different duplicate measurements were performed for all the other mutants. The rate was measured as slope of the 3rd overtone and averaged among the multiple injections. The data are presented as average values with error bars indicating the standard deviation.

**Fibril dissociation at pH 7.4**. Fibril dissociation at pH 7.4 was probed by measuring ThT and intrinsic fluorescence change over time in two series of triplicates. The ThT measurements were performed by mixing 40 μl of 100 μM PI3K-SH3 fibrils and 50 μM ThT in 10 mM glycine-hydrochloride pH 2 with 60 μl of 100 mM sodium phosphate pH 7.4. The mixing was carried out using the injection system of a CLARIOstar plate reader (BMG LABTECH, Ortenberg, Germany) and measuring immediately afterwards by exciting at 440 nm and recording the signal intensity at 480 nm. The intrinsic tryptophan fluorescence measurements were carried out by mixing the same two solutions (without ThT) by pipetting, followed by the measurement of fluorescence spectra every 15 s by exciting at 290 nm and recording between 300 and 380 nm in 2 nm intervals.

The analysis of soluble peptide by concentration determination at the end of the dissolution experiment was performed after one night of equilibration after the mixing of the two solutions mentioned above (without ThT). The samples were spun down for 30 min at 16,100 g. The protein concentration in the supernatant was determined by measuring the asorbance at 280 nm together with the extinction coefficient of PI3K-SH3 of $\varepsilon_{280} = 15{,}930$ M⁻¹cm⁻¹ using a V650 UV-Vis spectrophotometer (Jasco, Easton, MD, USA).

**Negative stain and cryo-EM image acquisition**. Negatively stained fibrils were prepared on 400 mesh carbon-coated copper grids (S160-4, Plano GmbH, Germany), stained with 1% uranyl acetate, and imaged using a Libra120 electron microscope (Zeiss) operated at 120 kV. Cryo-preparation was performed on glow-discharged holey carbon films (Quantifoil R 1.2/1.3, 300 mesh). The sample containing 50 μM PI3K-SH3 was 4×/10×/20x diluted with 10 mM glycine-hydrochloride (pH 2) to a final concentration of 12.5, 5 or 2.5 μM monomer equivalent. A total sample volume of 2.5 μl was applied onto the carbon grid and blotted for 3.5 s before being cryo-plunged using a Vitrobot (FEI). With 110,000-fold nominal magnification 622 micrographs have been recorded on a Tecnai Arctica electron microscope operating at 200 kV with a field emission gun using a Falcon III (FEI) direct electron detector in electron counting mode directed by EPU data collection software. Each micrograph was composed of 60 fractions. Each fraction contained 42 frames, i.e. in total 2520 frames were recorded per micrograph. The samples were exposed for 65 s to an integrated flux of 0.4 e⁻/Å²/s. Applied underfocus values ranged between 1.5 and 2.25 μm. The pixel size was 0.935 Å, as calibrated using gold diffraction rings within the powerspectra of a cross grating grid (EMS, Hatfield).

**Cryo-EM image processing and helical reconstruction**. MotionCor2[55] was used for movie correction. Fitting CTF parameters for all 622 micrographs was performed using CTFFIND4[56]. Further image processing and 3D reconstructions were done with RELION-2[39,40]. Selection of 256 micrographs was done with CCTFFIND by estimating the maximum resolution at which Thon rings could be detected to be better than 5 Å. From these micrographs, 4540 fibrils were manually picked. From these fibrils, 103,733 segments were extracted using an overlap of 90 % between neighbouring segments. The size of the segment images is 220 pixels. For data set characterisation we performed 2D classification (Supplementary Fig. 11). As an initial model for the refinement we used a noise-filled cylinder.

After several rounds of 3D refinements with helical symmetry search, we found a problem with the tilt priors: the tilt angle distribution became bimodal with maxima at 85° and 95°. However, we would expect the tilt angles to show a unimodal distribution around 90°. The ~4.7 Å cross-β pattern is a strong signal and substantially affects the alignment. If the helical rise parameter is slightly smaller than the correct value, the cross-β pattern can still be aligned by changing the tilt angles to higher or lower values (which accordingly reduces the spacing of the cross-β pattern). To overcome this problem, we fixed the tilt prior to 90° by usage of the RELION option *helical_keep_tilt_prior_fixed*, and then first optimized the helical parameters. In subsequent refinements, the helical symmetry parameters were fixed and the tilt angles (together with the other angles) were optimized.

Since the automated 3D refinement in RELION did not yield high-resolution reconstructions, we performed gold-standard refinements by splitting the data into an even and an odd set by selecting entire fibrils (not just segments, as they are overlapping). The FSC curve (Supplementary Fig. 12) was computed between the two half-maps and yields a resolution (with the 0.143 criterion) of 3.4 Å.

The handedness of the fibril structure was determined by comparing the reconstructed density with AFM images (Supplementary Fig. 5). For this comparison, the 3D density map of the fibril was converted to a height profile using Chimera[57] as follows: set surface color by height and set the color scale to gray. Then set camera projection mode to orthographic and save the image. The alignment of the AFM image with the calculated height profiles yields a cross-correlation coefficient of 0.943 for the left-handed and 0.914 for the right-handed helix.

**Model building and refinement**. A single chain atomic model of PI3K-SH3 was built with Coot[58,59]. Subsequently, seven copies of a single chain were placed into the EM density map. At residues Gln[46] and Glu[47], between strands β4 and β5, the density map is slightly ambiguous and could possibly be in agreement with an alternative interpretation for the trace of the Cα-chain (Supplementary Fig. 13).

The final model containing seven helical symmetry-related chains was used for further real space refinement in PHENIX[60]. Refinement was carried out using a resolution cut-off of 3.4 Å and NCS restraints between all seven subunits. At later stages of the refinement, hydrogen-bond restraints were defined for the cross-β sheets and Ramachandran restraints were used. The model-map FSC curve as obtained from *phenix.real_space_refine* is shown in Supplementary Fig. 12 (dashed line). The final statistics on the details of the refinement are shown in Supplementary Table 1. Molecular graphics and analyses were performed with Chimera[57].

**Reporting summary**. Further information on research design is available in the Nature Research Reporting Summary linked to this article.

## Data availability

The structure of the PI3K-SH3 fibril has been deposited in the Protein Data Bank under accession code 6R4R [https://doi.org/10.2210/pdb6R4R/pdb]. The 3.4 Å cryo-EM density map has been deposited in the Electron Microscopy Data Bank under accession code EMD-4727. The source data underlying Fig. 6b and Supplementary Fig. 8 are provided as a Source Data file. Other data are available from the corresponding authors upon reasonable request.

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

## Acknowledgements

We thank P.J. Peters and C. López-Iglesias for their advice and helpful discussions, H. Duimel for help with sample preparation, and the M4I Division of Nanoscopy of Maastricht University for microscope access and support. We thank P. Neudecker for helpful comments on the paper. We gratefully acknowledge the computing time granted by the Jülich Aachen Research Alliance High-Peformance Computing (JARA-HPC) Vergabegremium and VSR commission on the supercomputer JURECA at Forschungszentrum Jülich. Computational support and infrastructure was provided by the Center for Information and Media Technology (ZIM) at the University of Düsseldorf (Germany). A.K.B. and N.V. thank the Deutsche Forschungsgemeinschaft (DFG) for funding. D.W. was supported by grants from the Portfolio Technology and Medicine, the Portfolio Drug Design, and the Helmholtz-Validierungsfonds der Impuls und Vernetzungs-Fonds der Helmholtzgemeinschaft. This study was funded in part by the DFG SFB 974 and SFB 1208 (to D.W.). Support from a European Research Council (ERC) Consolidator Grant (grant agreement no. 726368) to W.H. is acknowledged. A.K.B. thanks the Novo Nordisk Foundation for support through a Novo Nordisk Foundation Professorship.

## Author contributions

A.K.B. and G.F.S. conceived the study. N.V. and L.N.M. performed the biochemical experiments. A.K.B., N.V., L.N.M. analysed the biochemical and kinetics data. R.G.B.R. performed cryo-EM experiments and initial data analysis. C.R. and G.F.S. performed image processing, reconstruction and model building. C.R., A.K.B., N.V., G.F.S., W.H. and L.G. wrote the paper. D.W. and all other authors discussed results and commented on the paper.

## Additional information

**Competing interests:** The authors declare no competing interests.

