## [Peer Review File · Nature Communications]

Reviewers' comments:

Reviewer #1 (Remarks to the Author):

The manuscript by Röder et al. reports cryo-EM investigations of the molecular/atomic structure of amyloid fibrils formed by the protein PI3K-SH3. The derived model reveals a complex, staggered monomer arrangement in the straight filaments, which further interact with each other to form twisted double-filament fibrils. The interface between the two straight filaments turns out to be surprisingly large, despite supporting AFM measurements indicating only weak inter-filament interactions. Finally, the effect of selected point mutations on aggregation rate is investigated by QCM-D and explained in the context of the derived model. The authors further extend this discussion to previously investigated mutations in order to rationalize the corresponding alterations in aggregation behavior reported in literature.

While more and more cryo-EM structures of medically relevant amyloid fibrils formed by various proteins are nowadays reported in literature, the authors here derive an atomic model of a classic model system that has been studied extensively during the last decades. Despite this large body of work, many observations such as the various effects of the numerous point mutations studied could so far be explained only in a somewhat hand-waving manner. The work by Röder et al. now makes it possible to rationalize many of the previously obtained results and promises the development of a consistent picture of the aggregation of this important model system. This manuscript thus represents a very significant and important work. The paper is very well written, the experiments appear sound, and the conclusions mostly justified.

My only concerns are related to the AFM and QCM-D measurements of the aggregation of the different mutants. In Fig. 6b and on page 12, the relative rates of aggregation of the different mutants are given and compared to the AFM images in Fig. S9. I have some doubts whether the correlation between AFM and QCM-D is as strong as the corresponding discussion describes it. In particular, the measured rates fall into two classes: I77A and I82A have rates between 1 and 2, while I22A, I29A, and I53A have rates between 0.01 and 0.02. Therefore, I'd expect mutants with similar rates also displaying similar aggregation behaviors in the AFM investigations. However, this is clearly not the case, as Fig. S9 reveals serious and very similar fibril formation for all mutants (and wt) except I53A. Therefore, in AFM, the two mutants with rates of 0.02 behave almost identical to the ones with rates around 1, while I53A with a rate of 0.01 shows almost no aggregation. How can this be explained? In the experimental section, it is further mentioned that all QCM-D measurements were performed twice, with the exception of I53A, which was measured only once. This casts even more doubt on the correlation between aggregation rates and AFM. The authors should thus repeat the QCM-D and AFM measurements for the I53A mutant at least once and verify that the observed behaviors are no artefacts. Assuming that the reported results are reproducible, it is furthermore important to assess whether the observed small fibrils in Fig. S9d are actually assembled from the mutant or just represent the wt seeds. Therefore, please provide a reference image of the seeds only. Furthermore, I'd appreciate if the authors could show the QCM-D results (ΔF and ΔD curves) for all mutants in the SI to enable a direct comparison.

Once these issues have been addressed, I expect the paper to be ready for publication in Nature Communications.

Reviewer #2 (Remarks to the Author):

Röder et al. present the 3.4Å cryo-EM structure of the PI3K-SH3 amyloid fibril. Model building allowed to determine the position of the first 77 residues out of 86 amino acids. SH3 domains are found in a large number of other proteins, where they can contribute to protein-protein interactions.

The work is of solid quality. The obtained resolution is excellent, given that an Arctica microscope was used, while a Titan would possibly have given higher resolution. The final map is of high resolution and excellent quality, allowing clear definition of the atomic fold of the proteins in the

fibril, with the exception of the likely non-folded C-terminus. The manuscript is well written, and the figures are clear and not redundant.

Minor comments are:

Page 5, Results: The title says Cryo-EM, but the first thing to be mentioned and shown in Figure 1 is AFM data. This should correspond to each other. Show cryo-EM here, and AFM in Suppl. Mat., or correct the title to "Structural Analysis by AFM and Cryo-EM"

Page 11: "We probed the sensitivity ... by substituting isoleucine residues for alanine across the protein sequence." This was done by expression and purification, in vitro, and not in silico? Please state clearly.

Page 18: Magnification is specified as 110'000x. The detector is given as a Falcon III, which has a pixel size of 14 micrometers. That would correspond to a pixel size at the sample level of 1.27A/px. However, the authors state a pixel size as 0.935A. This means that a magnification difference of 1.358x is present. Where does this come from? What is the definition of the "magnification"? This microscope doesn't have a viewing screen for an operator, only a web-cam screen, and then the Falcon III. It would make more sense to either omit the specification of the magnification, or state it as 150'000x. Same for Table 1.

Page 19: "A maximum resolution of better than 5A was estimated by CTFFIND for 256 micrographs, from which ..." This is referring to the CTF Thon-ring fit resolution, not the resolution of the micrographs or the structural information included in it. Please use a more precise wording, as this could be misleading for people not familiar with CTFFIND.

Table 1: A helical screw with approximate 2sub1 symmetry is specified, but the fibril is still classified as C1. It would be useful to add that this has a pseudo 2-sub-1 helical symmetry, since some amyloid filaments don't have that and are cleanly C2 or purely C1.

Figure S4: Comparison with Jimenez et al. (1999): 20 years ago, people didn't have today's cameras or algorithms, and mostly did negative stain EM. I don't think the reader gains anything from comparing a 3.4A structure with a 20-year old negative stain analysis. That Jimenez et al. (1999) work is still beautiful for its time, and showed the different strains already then. This comparison of the structure from 1999 with today's map doesn't improve the present work. I recommend omitting this figure, but still citing the old work.

Figure S7: These small windows on the interface are not really revealing anything interpretable. For example, these thumbnail images don't allow to understand, if there are additional contacts between two strands outside of the field of view. Please increase field of view, or omit.

Figure S11: Interesting fit to FSC curve with $1/(1+\exp(x-A)/B)$. What are the fitted parameters A and B? Please indicate with a vertical line, where the resolution limit 3.4A would be. Please indicate with a dotted line the FSC=0.143 threshold, and with another horizontal line the FSC=0 line, so that readers can see, if and how far the FSC curve approaches zero at high resolution.

Reviewer #3 (Remarks to the Author):

The manuscript "Atomic Structure of PI3-Kinase SH3 Amyloid Fibrils by Cryo- Electron Microscopy" by Röder et al describes the Cryo-EM structure of fibrils of the SH3 domain from PI3-Kinase, a widely used model for the study of amyloid aggregation. The manuscript is sound, clear and well written. Here below, some comments follow:

- It is very hard to picture the difference of a "half of a beta-stagger" between the two

protofilaments from Fig. 2C. Authors may try some transparent stripes, lines or other ways to show that corresponding secondary structures are not at the same z-position.

- In Fig. 5 consecutive staggers are numbered i , $i+2$, $i+4$... even though that the Authors have already successfully used this way to number staggers, it would be useful to define in the legend (or elsewhere) that 1=half-stagger and then that $i+2$ is indicating the molecule adjacent to molecule i .

- At page 14 Authors discuss "the transition state of the elongation reaction". How do they really define it? Given that their fibril elongation occurs under denaturing conditions it is rather straightforward to define it but I believe that for sake of clarity it would be useful a clarifying sentence. Under these denaturing conditions, they can ignore possible effects of the mutations on the native state, however, can they completely exclude that the effects observed on the amyloidogenicity do not come from different conformation(s) of the unfolded protein?

- At page 15 the Authors discuss some mutants from the literature (K16Q, E52K, E61K). In their discussion they seem to exclude that mutations may trigger a re-arrangement of the polypeptide resulting in a different fibrillar structure. The examples of tau or of light chains (even though of different sequence) would point to very strong plasticity in the tertiary structure.

- In the core of this fibril there is a remarkable number of acidic residues. Given that SH3 is a model system, the question of the physiological significance of amyloid aggregates grown at pH 2 is totally negligible. However, now that few fibrillar structures are available, it would enrich the discussion if the Authors would take the opportunity to analyse also other Cryo-EM structures available and check if acid residues are found in non-solvent exposed positions only in fibrils grown at low pH (in b2m ones there are two) or if it is a general trait.

- In the same line of thoughts, once formed are the SH3 fibrils stable at physiologic pH? Or do they need to be kept at low pH all along?

- Did the Authors check if, by any chance, the polypeptide undergoes a spontaneous proteolysis resulting in the loss of the flexible C-terminal stretch once the fibrils are formed? Especially in case it does, this observation would add up in the current debate in the field over proteolysis as trigger of amyloid aggregation or a phenomenon occurring after amyloids are formed to unstructured and flexible regions.

Reviewers' comments:

Reviewer #1 (Remarks to the Author):

The manuscript by Röder et al. reports cryo-EM investigations of the molecular/atomic structure of amyloid fibrils formed by the protein PI3K-SH3. The derived model reveals a complex, staggered monomer arrangement in the straight filaments, which further interact with each other to form twisted double-filament fibrils. The interface between the two straight filaments turns out to be surprisingly large, despite supporting AFM measurements indicating only weak inter-filament interactions. Finally, the effect of selected point mutations on aggregation rate is investigated by QCM-D and explained in the context of the derived model. The authors further extend this discussion to previously investigated mutations in order to rationalize the corresponding alterations in aggregation behavior reported in literature.

While more and more cryo-EM structures of medically relevant amyloid fibrils formed by various proteins are nowadays reported in literature, the authors here derive an atomic model of a classic model system that has been studied extensively during the last decades. Despite this large body of work, many observations such as the various effects of the numerous point mutations studied could so far be explained only in a somewhat hand-waving manner. The work by Röder et al. now makes it possible to rationalize many of the previously obtained results and promises the development of a consistent picture of the aggregation of this important model system. This manuscript thus represents a very significant and important work. The paper is very well written, the experiments appear sound, and the conclusions mostly justified.

My only concerns are related to the AFM and QCM-D measurements of the aggregation of the different mutants. In Fig. 6b and on page 12, the relative rates of aggregation of the different mutants are given and compared to the AFM images in Fig. S9. I have some doubts whether the correlation between AFM and QCM-D is as strong as the corresponding discussion describes it. In particular, the measured rates fall into two classes: I77A and I82A have rates between 1 and 2, while I22A, I29A, and I53A have rates between 0.01 and 0.02. Therefore, I'd expect mutants with similar rates also displaying similar aggregation behaviors in the AFM investigations. However, this is clearly not the case, as Fig. S9 reveals serious and very similar fibril formation for all mutants (and wt) except I53A. Therefore, in AFM, the two mutants with rates of 0.02 behave almost identical to the ones with rates around 1, while I53A with a rate of 0.01 shows almost no aggregation. How can this be explained? In the experimental section, it is further mentioned that all QCM-D measurements were performed twice, with the exception of I53A, which was measured only once. This casts even more doubt on the correlation between aggregation rates and AFM. The authors should thus repeat the QCM-D and AFM measurements for the I53A mutant at least once and verify that the observed behaviors are no artefacts. Assuming that the reported results are reproducible, it is furthermore important to assess whether the observed small fibrils in Fig. S9d are actually assembled from the mutant or just represent the wt seeds. Therefore, please provide a reference image of the seeds only. Furthermore, I'd appreciate if the authors could show the QCM-D results (ΔF and ΔD curves) for all mutants in the SI to enable a direct comparison.

We thank the reviewer for the overall very positive evaluation of our study.

The AFM images were not originally intended to be quantitatively analysed, but rather to illustrate the morphology of the grown fibrils. Our analysis of the relative elongation rates is based on the QCM experiments, which is a much more appropriate and accurate technique for the measurement of relative elongation rates.

Given that the AFM images have been taken after two days of incubation of the seeds with the different sequence variants, we do not expect the length of the fibrils to scale with the elongation rate. The lengths of the fibrils at the end of the reaction, i.e. when the monomer concentration has reached its equilibrium value, is mostly determined by the total amount of monomer that had been added, rather than by the elongation rate. Nevertheless, we agree with the referee that it is remarkable that the seeds that had been incubated with I53A are much shorter after two days than in all the other cases. This result hints towards the elongation rate of I53A being significantly lower than that of I22A and I29A. We therefore have, as suggested by the reviewer, performed several more repeats of the QCM elongation experiments with I53A and we have updated the figure with the new results. We find indeed that the WT seeds grow significantly more slowly (approximately one order of magnitude) when exposed to a given concentration of I53A compared to when exposed to the same concentration of I22A or I29A. Therefore, the QCM data are compatible with the very slow elongation rate observed for the I53A mutant. Indeed, the high sensitivity of QCM enables to resolve the fibril elongation by I53A.

We have added raw data of the QCM experiments in the supplementary material as requested by the reviewer (Fig. S9) as well as an AFM picture for the WT seeds as a reference (Fig. S10).

Once these issues have been addressed, I expect the paper to be ready for publication in Nature Communications.

Reviewer #2 (Remarks to the Author):

Röder et al. present the 3.4Å cryo-EM structure of the PI3K-SH3 amyloid fibril. Model building allowed to determine the position of the first 77 residues out of 86 amino acids. SH3 domains are found in a large number of other proteins, where they can contribute to protein-protein interactions.

The work is of solid quality. The obtained resolution is excellent, given that an Arctica microscope was used, while a Titan would possibly have given higher resolution. The final map is of high resolution and excellent quality, allowing clear definition of the atomic fold of the proteins in the fibril, with the exception of the likely non-folded C-terminus. The manuscript is well written, and the figures are clear and not redundant.

Minor comments are:

Page 5, Results: The title says Cryo-EM, but the first thing to be mentioned and shown in Figure 1 is AFM data. This should correspond to each other. Show cryo-EM here, and AFM in Suppl. Mat., or correct the title to "Structural Analysis by AFM and Cryo-EM"

We now refer to old Fig.2 earlier in the manuscript and present now old Fig 2 as new Fig 1 to avoid this mismatch.

Page 11: "We probed the sensitivity ... by substituting isoleucine residues for alanine across the protein sequence." This was done by expression and purification, in vitro, and not in silico? Please state clearly.

We have now stated more clearly in the text that we have produced these sequence variants and carried out the experiments ("We expressed and purified the sequence variants and experimentally quantified the rates").

Page 18: Magnification is specified as 110'000x. The detector is given as a Falcon III, which has a pixel size of 14 micrometers. That would correspond to a pixel size at the sample level of 1.27Å/px. However, the authors state a pixel size as 0.935Å. This means that a magnification difference of 1.358x is present. Where does this come from? What is the definition of the "magnification"? This microscope doesn't have a viewing screen for an operator, only a web-cam screen, and then the Falcon III. It would make more sense to either omit the specification of the magnification, or state it as 150'000x. Same for Table 1.

This is good point and we thank the reviewer for pointing this out. We now use "nominal magnification" instead of simply "magnification", which refers to the magnification as printed on the objective lens. The reviewer is correct about the postmagnification factor of 1.358 for the Falcon III. We actually have, apart from webcam screen, three detectors on our microscope, each with its own postmagnification factor.

Page 19: "A maximum resolution of better than 5Å was estimated by CTFFIND for 256 micrographs, from which ..." This is referring to the CTF Thon-ring fit resolution, not the resolution of the micrographs or the structural information included in it. Please use a more precise wording, as this could be misleading for people not familiar with CTFFIND.

We agree with the reviewers's comment. We rephrased the sentence to: "Selection of 256 micrographs was done with CCTFFIND by estimating the maximum resolution at which Thon rings could be detected to be better than 5 Å."

Table 1: A helical screw with approximate 2sub1 symmetry is specified, but the fibril is still classified as C1. It would be useful to add that this has a pseudo 2-sub-1 helical symmetry, since some amyloid filaments don't have that and are cleanly C2 or purely C1.

We agree, the actual symmetry was unclear from Table 1. We therefore added to Table 1 that the structure has a pseudo 2₁ symmetry.

Figure S4: Comparison with Jimenez et al. (1999): 20 years ago, people didn't have today's cameras or algorithms, and mostly did negative stain EM. I don't think the reader gains anything from comparing a 3.4Å structure with a 20-year old negative stain analysis. That Jimenez et al. (1999) work is still beautiful for its time, and showed the different strains already then. This comparison of the structure from 1999 with today's map doesn't improve the present work. I recommend omitting this figure, but still citing the old work.

The purpose of this comparison was not to demonstrate that we could improve on a 20-year-old structure, which is, given the improved detectors and microscopes, not too much of an achievement, but to show how similar their structure is to ours. We think it is still interesting to show that Jimenez et al. very likely studied the same types of fibrils. We tried to formulate this comparison in the figure caption more positively, as we of course do not want to lessen their achievement.

The caption to Fig. S4 now reads:

"Overlay of the presented DF PI3K-SH3 fibril model with the low-resolution cryo-EM density (contour graphically extracted and mirrored) from Jimenez et al. (1999)⁵. The model and the density are in good agreement, which suggests that both preparations likely yield the same structure."

Furthermore, we realized that the mirrored density from Jimenez et al. fits even a bit better and therefore we now changed the handedness of the density in Fig. S4.

Figure S7: These small windows on the interface are not really revealing anything interpretable. For example, these thumbnail images don't allow to understand, if there are additional contacts between two strands outside of the field of view. Please increase field of view, or omit.

We increased the field of view and now show the entire cross-sections, which we think improves the clarity of the figure (Fig. S7).

Figure S11: Interesting fit to FSC curve with $1/(1+\exp(x-A)/B)$. What are the fitted parameters A and B? Please indicate with a vertical line, where the resolution limit 3.4Å would be. Please indicate with a dotted line the FSC=0.143 threshold, and with another horizontal line the FSC=0 line, so that readers can see, if and how far the FSC curve approaches zero at high resolution.

The fit parameters are $A=0.2325$ and $B=0.03286$. We added these values to the caption of Figure S11. In addition we also added the 0.143 and zero lines and the vertical 3.4 Å line.

Reviewer #3 (Remarks to the Author):

The manuscript “Atomic Structure of PI3-Kinase SH3 Amyloid Fibrils by Cryo- Electron Microscopy” by Röder et al describes the Cryo-EM structure of fibrils of the SH3 domain from PI3-Kinase, a widely used model for the study of amyloid aggregation. The manuscript is sound, clear and well written. Here below, some comments follow:

- It is very hard to picture the difference of a “half of a beta-stagger” between the two protofilaments from Fig. 2C. Authors may try some transparent stripes, lines or other ways to show that corresponding secondary structures are not at the same z-position.

We added horizontal lines for Fig. 1C (previously Fig. 2C) to show the staggering of opposite layers and added a label to indicate the 2.35 Å shift.

- In Fig. 5 consecutive staggers are numbered i , $i+2$, $i+4$... even though that the Authors have already successfully used this way to number staggers, it would be useful to define in the legend (or elsewhere) that 1 =half-stagger and then that $i+2$ is indicating the molecule adjacent to molecule i .

We have now explained in the figure legend more extensively the numbering convention, including that i and $i+2$ are adjacent layers, etc.

- At page 14 Authors discuss “the transition state of the elongation reaction”. How do they really define it? Given that their fibril elongation occurs under denaturing conditions it is rather straightforward to define it but I believe that for sake of clarity it would be useful a clarifying sentence. Under these denaturing conditions, they can ignore possible effects of the mutations on the native state, however, can they completely exclude that the effects observed on the amyloidogenicity do not come from different conformation(s) of the unfolded protein?

With "transition state of the elongation reaction", we simply refer to the structure or structural ensemble at the top of the highest free energy barrier between isolated monomer and incorporated monomer. Based on the data in this manuscript, we cannot say anything about how closely the transition state resembles the initial, monomeric state or the final, fibrillar state. The changes in elongation rate that we observe simply mean that the free energy difference between the starting point and the top of the barrier changes, and we cannot at this point exclude that this is partly caused by a change in the energy landscape of the unfolded monomer ensemble. We have reformulated the text accordingly, now at page 15.

- At page 15 the Authors discuss some mutants from the literature (K16Q, E52K, E61K). In their discussion they seem to exclude that mutations may trigger a re-arrangement of the polypeptide resulting in a different fibrillar structure. The examples of tau or of light chains (even though of different sequence) would point to very strong plasticity in the tertiary structure.

All experiments, including the ones from the literature (K16Q, E52K, E61K) were performed by using WT seeds. In this type of experimental setup, growth of the seeds is only observed if the monomeric sequence variant is able to adopt the structure of the

WT seed. The overall setup (strongly seeded) and time scales involved in the QCM experiments ensures that only fibril growth is measured, no de novo formation of fibrils. This is confirmed by QCM data without surface-attached WT seeds, where no signal is observed. Therefore, we are confident that we can interpret all the kinetic data of elongation by mutants by means of the WT fibril structure. We added a paragraph on what has now become pages 15 and 16.

- In the core of this fibril there is a remarkable number of acidic residues. Given that SH3 is a model system, the question of the physiological significance of amyloid aggregates grown at pH 2 is totally negligible. However, now that few fibrillar structures are available, it would enrich the discussion if the Authors would take the opportunity to analyse also other Cryo-EM structures available and check if acid residues are found in non-solvent exposed positions only in fibrils grown at low pH (in b2m ones there are two) or if it is a general trait.

*Thank you for pointing this out. It is indeed a very intriguing thought. Nevertheless, we could find acidic residues in both pH2 **and** non-acidic pH-derived fibrils (e.g. brain extracted murine AA amyloid shows 4 internal, acidic residues per monomer). It appears that the internal placement of acidic residues does not solely apply for fibrils that are grown at low pH. However, the protonation state at pH 7 seemingly contributes to the (in-)stability of our fibril (c.f. next comment and page 11 in the manuscript where we added: "Further hints towards the role of electrostatic interactions role in maintaining the structure, comes from a pH-shift experiment. By changing the pH from 2 to 7.4, we observed that fibrils depolymerise almost completely after 1 hour (Fig. S8). We propose three clusters that may be highly influenced by this pH shift: a) the space between strands $\beta 5$ and $\beta 6$ (Asp⁴⁴(i), Arg⁶⁶(i-2), Glu⁴⁷(i)); b) the interactions between Asp¹³(i) and Lys¹⁵(i), and between Glu¹⁷(i+4) and Asn⁵⁷(i); c) the solvent-exposed and protonatable patch involving Asp²¹(i), Asp²³(i) and His²⁵(i) (also highlighted as having a fundamental role in amyloid formation).")*

- In the same line of thoughts, once formed are the SH3 fibrils stable at physiologic pH? Or do they need to be kept at low pH all along?

We tested whether the fibrils are stable if brought to neutral pH and we found that they dissolve. Measurements of the supernatant concentration after mixing 40 μ l of 100 μ M fibrils in 10 mM glycine-HCl with 60 μ l of 100 mM sodium phosphate buffer pH 7.4, incubation for 24 hours and centrifugation for 30 min at 16,100 g, are shown in the Figure S8 (right panel) as percentage of the total protein concentration (40 μ M). We also monitored the dissociation kinetics by intrinsic fluorescence and ThT fluorescence, mixing the above mentioned solutions. The results are shown in the left panel of Fig. S8. Further description of the method was added in the chapter "Fibril dissociation at pH 7" of the Material and Methods section (page 19), and a description (see answer to previous point) was added in the "Architecture of the PI3K-SH3 Amyloid Fibril" chapter of the Results section (page 11).

- Did the Authors check if, by any chance, the polypeptide undergoes a spontaneous proteolysis resulting in the loss of the flexible C-terminal stretch once the fibrils are formed? Especially in case it does, this observation would add up in the current debate

in the field over proteolysis as trigger of amyloid aggregation or a phenomenon occurring after amyloids are formed to unstructured and flexible regions.

We tested for proteolytic cleavage at pH 2 of the fibrils. The same sample of fibrils analysed by cryo-EM (Fibrils) was tested one year after its preparation and it showed no cleavage compared to freshly diluted monomers (Mon.). To prepare the SDS-Page, the fibrils were diluted in pH7 to depolymerise them, then the concentration of the sample was adjusted to 20 μ M and compared to an equally concentrated sample of monomeric protein freshly dissolved in pH 7. The samples were both heated at 95°C with the loading buffer for 5 minutes.

REVIEWERS' COMMENTS:

Reviewer #1 (Remarks to the Author):

The authors have addressed all my concerns. I recommend publication of the paper in Nature Communications.

Reviewer #3 (Remarks to the Author):

The Authors have satisfactorily addressed my concerned and I recommend this manuscript for publication.

I found a typo at pg 5 "cyro-EM" instead of Cryo-EM.